# Hydrodynamic and morphodynamic patterns on a mid-channel intertidal bar in an estuary

Marco Schrijver<sup>1,2</sup>, Maarten van der Vegt<sup>1</sup>, Gerben Ruessink<sup>1</sup>, and Maarten G. Kleinhans<sup>1</sup>

<sup>1</sup>Department of Physical Geography, Faculty of Geosciences, Utrecht University, Princetonlaan 8a, 3584 CB Utrecht, the Netherlands

<sup>2</sup>Rijkswaterstaat Zee en Delta, Poelendaelesingel 18, 4335 JA Middelburg, the Netherlands

**Correspondence:** Marco Schrijver (m.c.schrijver@uu.nl)

**Abstract.** Mid-channel bars that emerge during low tide are a common feature of estuaries. In contrast to fringing tidal flats, these bars are fully surrounded by channels, which may lead to different hydrodynamics and sediment dynamics. Exactly how is unclear, and whether simple models for the dynamics of fringing flats are applicable to mid-channel bars is likewise unknown. These insights are needed for large-scale morphological prediction of the effects of human interference and sea level rise. By using a unique dataset consisting of six months of current velocity data at 16 locations on a mid-channel bar, we analysed the hydrodynamics and calculated a proxy for sediment transport. Taking advantage of the large number of measurement locations, detailed flow patterns were derived. The data show that the spatially non-uniform morphology influences the hydrodynamics and sediment transport in multiple ways. At the deeper parts of the mid-channel bar the ebb- or flood-dominance is determined by the surrounding ebb- and flood channels. This causes changes in ebb or flood dominance over short distances. On the tidal flat itself, a higher area on the seaward bar head causes large-scale circulation cells. Here the flow bends around the emerged areas and causes larger cross-shore flows than expected for alongshore uniform tidal flats. These large-scale flow patterns also determine sediment transport patterns and possibly explain how the mid-channel bar can increase in height. Although the measured data show that velocity can veer over depth, this does not result in large changes in sediment transport direction. This suggests that depth-averaged models are reasonable approximations for predicting sediment transport, sedimentation and erosion on mid-channel bars. Nonetheless, the effects of veering on long-term morphological development of intertidal areas remain to be assessed.

### 1 Introduction

Intertidal areas in estuaries and coastal lagoons are a critical habitat for many benthic and bird species and provide ecosystem services such as coastal protection, wildlife conservation, economic activities, and other social benefits (Costanza et al., 1997; Dyer and Huntley, 1999; Pinto et al., 2010; Barbier et al., 2011; Coco et al., 2013; Plag and Jules-Plag, 2013). The morphology of intertidal areas plays a crucial role in sustaining ecosystem functioning, as it determines, along with the imposed variations in water level due to tides and storms, the exposure time and the total wetted area. Human use, invasive species, sea level rise, and climate change pose threats to the long-term survival of the intertidal areas and their ecosystem services (Kennish, 2002; Leuven and Kleinhans, 2019; de Vet et al., 2020; van Dijk et al., 2021; Xie et al., 2022). In order to preserve the

ecosystem services of intertidal areas, their morphology needs to change in response to changes in local sea level conditions. To predict these morphological changes, knowledge is required of the sediment transport patterns in intertidal areas in relation to the hydrodynamic conditions (tidal range, current flow, and wave exposure), sediment composition, and human activities. Additionally, the question is whether frequently used 1D or 2D models sufficiently predict these changes.

30

Intertidal areas with different hydrodynamic settings can be found in a wide range of environments. For sediment transport patterns, insight into the main directions of tidal flow and wave exposure are important (Dyer, 1989; van Rijn, 1993). On the landward side of the tidal basins and near barrier islands, we find intertidal areas where the tide-induced flow velocities occur mainly across the tidal flats. The shape and morphological evolution of these systems, also known as open tidal flats, are predominantly determined by a balance between wave-driven processes that tend to erode the tidal flats and tides that transport sediment onto them (Hir et al., 2000; Roberts et al., 2000; Friedrichs, 2009). One-dimensional (1D) models and concepts have been used to gain understanding of how these flats develop as a function of changes in external forcing, sediment supply (Bearman et al., 2010), and direct human interventions such as building dikes and land reclamation (Gregory Hood, 2004). Given that the main trends in flow, sediment transport and morphology are in the cross-flat direction, the 1D-assumption that the tidal flat is uniform in the alongshore direction is considered a reasonable approximation.

In many estuaries we also find fringing tidal flats. These are located between a tidal channel and the bank of the estuary. Here, the alongshore tidal flows are much larger (Hanssen et al., 2024) and the morphological evolution is determined by the alongshore and cross-shore tidal flows, wind-driven flows, and waves. For these systems, the tidal flats are often assumed to be alongshore uniform, resulting in the use of 1D models and concepts (Waeles et al., 2004; Maan et al., 2015; Zhou et al., 2016). However, the estuary width often fluctuates on top of a landward decreasing trend, so that the width and area of the fringing tidal flats are not constant (Leuven et al., 2017). Hence, alongshore variability in the morphology of fringing tidal flats can be expected, but how this influences the local alongshore and cross-shore tidal flows and the morphological evolution has not yet been studied in detail.

A third type of intertidal area is found in the form of mid-channel bars, typically situated in estuaries, but also in coastal lagoons. These mid-channel bars differ from fringing tidal flats in the sense that they are bounded by two channels instead of one. These channels form ebb and flood dominant channels (van Veen, 1950; Jeuken, 2000; Leuven et al., 2016), in which one of the channels may have net sediment transport in the ebb direction and the other in the flood direction, often resulting in a tendency for sediment to recirculate around a mid-channel bar. Furthermore, the mid-channel bars have a finite length and are therefore by definition not alongshore uniform. The flow must curve around the heads of the bar, which may cause upslope veering of the near-bed flow towards what is effectively an inner bend of the channel around the bar head. Such secondary flows may drive upslope sediment transport that is absent on commonly uniform fringing tidal flats. Another difference from fringing tidal flats is that a mid-channel bar is flooded and drained from multiple directions. Once fully submerged, tidal differences between channels may affect large-scale flow patterns. These differences with fringing tidal flats are likely to have implications for the hydrodynamics, sediment transport, and morphological evolution of a mid-channel bar.

While much understanding has been gained of flow and sediment transport on open and fringing tidal flats in estuaries, e.g., the Jade, and Weser estuary (Gelfort et al., 2011; Benninghoff and Winter, 2018; Lefebvre et al., 2021), San Pablo Way

(Allen et al., 2019), the Yangtze (Wan et al., 2014; Zhu et al., 2017), the Gironde estuary (Billy et al., 2012), the Seine estuary (Grasso and Le Hir, 2018), the Belon estuary (Michel et al., 2021), and Oosterschelde (de Vet et al., 2020), three-dimensional (3D) flow patterns on mid-channel bars are neither well known nor quantified. Measurements are sparse and costly, and morphodynamic models often use depth-averaged flow velocities (either 1D or 2DH) for reasons of computational efficiency to calculate sediment transport and morphological change, even though flow patterns may in fact be 3D. As a result, the patterns developing in 2D morphological models depend strongly on sediment transport parametrization for veering of the flow and the bed shear stress vector, and for slope effects on sediment transport that need to be calibrated to obtain realistically shaped channels and bars (Baar et al., 2019). Whether this 2D approach is a sufficiently valid approach for these mid-channel bars is unclear.

This study aims to increase our understanding of the hydro- and morphodynamics and their impact on the sediment transport on mid-channel bars. Moreover, we will compare the results for mid-channel bars conceptually with those of open and fringing tidal flats to gain insight into the differences between these three types of intertidal areas. For this, we analysed the current velocities on a spatial grid at 16 locations for a period of 6 months on a mid-channel tidal bar in the Western Scheldt estuary in the Netherlands. The results are compared to simple models developed for open and fringing flats, which serve as a starting point for understanding differences with mid-channel bars. From this comparison we will infer whether the use of 1D and 2D models sufficiently to predict morphological changes of an intertidal bar, or that more sophisticated 3D models are needed. The measured cross-shore and alongshore velocities were used as input into a simple sediment transport predictor to understand how the hydrodynamic forcing and tidal flat morphology determine the sediment balance of a mid-channel tidal bar. Although flow patterns on intertidal shoals have numerically been studied (de Vet et al., 2018; Elmilady et al., 2019, 2020), the specific effects of the non-uniform bathymetry was not elaborated in these studies. As the field site is macro-tidal and limited in fetch we assume that the effect of waves on the large-scale sediment transport patterns is small in comparison with the effect of currents. Therefore, the effect of waves on the sediment transport was omitted in this study.

### 2 Methods

# 2.1 Field site

The Western Scheldt, which is a part of the Scheldt estuary that is located in Belgium and the Netherlands (Meire et al., 2005), is a funnel shaped estuary and has a complex geometry containing mid-channel bars. The Western Scheldt is characterized by a dual channel system of evasive ebb and flood channels, fringing tidal flats, submerged mid-channel bars, mid-channel tidal bars such as 'Hooge Platen', sills, and short-cut channels that connect an ebb-dominant channel with a flood-dominated channel, see Fig. 1(a). Where ebb and flood channels meet, sills are present between the channel heads (van Veen, 1950). The ebb dominated channel is used as the main shipping channel to the ports of Flushing, Terneuzen, Ghent and Antwerp. One of the human interventions in the Western Scheldt is deepening and maintenance of the main shipping channel to sustain access to the ports. Therefore, approximately 10.5 Mm<sup>3</sup> sediment is dredged annually from the sills and deposited in designated areas in the Western Scheldt (van Dijk et al., 2021).

The field site is an intertidal flat, located in the northern part of the mid-channel bar 'Hooge Platen' in the Western Scheldt. A measurement campaign was carried out from February 2019 to July 2019. The field site has a semidiurnal tide with an average tidal range of 3.83 m, 4.45 m at spring tide and 2.99 m at neap tide (Rijkswaterstaat, 2018). The dominant wind direction in the area is south-west to west. As depicted in Fig. 1(a) this mid-channel tidal bar is surrounded by multiple channels: in the south and east ebb dominant channels, and in the west and north flood dominant channels. On the northeastern shore a spit separates the near-bar flow from the large adjacent channel in the east. The mid-channel bar is fully submerged during high tide, except for the area in the south-west, which is submerged only when the water level is above 3 m NAP (NAP = Dutch ordnance datum, approximately mean sea level). An elevation map of 'Hooge Platen' is shown in Fig. 1(b). The higher parts. i.e. the southern shore of the bar and 'de Bol' are overgrown. At other areas, the bed composition of the top layer consists of a mixture of sand and silt. In general, the bed is smooth, with local small ripples (1-2 cm). Several smaller creeks run from higher elevated areas towards the channel. Halfway the field site a large creek is present with a width of several meters and depth up to 1.5 m. The field site consists of a mixture of sand and finer sediments. The sediment characteristics in the top layer varies from  $D_{50}$  = 95  $\mu$ m in the south-east to  $D_{50}$  = 196  $\mu$ m in the north-west. The percentage of sediment fraction with a size smaller than 63  $\mu$ m varies from 22.8% in the south-east to 1.4% in the north-west. In Fig. 1(c) the difference in bed level between 2009 and 2019 is shown. The figure shows that the mid-channel tidal bar is locally elevating up to 10 - 15 cm per year, which is much faster than local sea level rise of 2 mm per year. Peak flow velocities decrease as the bed level becomes higher. As a result, finer sediments are more likely to settle, thereby increasing the percentage of silt in these areas.

## 110 2.2 Data gathering

100

115

120

125

A detailed overview of the field site including the position of the transects and the 16 sensors is shown in Fig. 2. At transects T1 to T4 velocity profiles were measured at four locations per transect: one subtidal location in the channel and three on the intertidal flat (-1 m NAP, 0 m NAP and +1 m NAP). In this paper, each measurement location is referenced by the one-digit transect number of its alongshore location from west to east, followed by the one-digit number of its cross-shore position on the transect: 1 is deepest, 4 is most elevated.

Velocity profiles in the channel were recorded during the period of 15 February 2019 to 5 July 5 2019 using upward looking Teledyne RD Instruments Workhorse Monitor 1200 kHz mounted at a height of 0.6 m in a frame that was placed at the sea bed. Each instrument was configured with a cell size of 0.5 m and a blanking distance of 1.04 m. Every 10 minutes the average of the preceding ten minutes data were stored. Bottom depth at the locations of the Teledyne RD Instruments Workhorse Monitor 1200 kHz was obtained through the sonar sensor mounted on the vessel during placement of the instruments. Note that location '41' had a less clear signal because it was positioned at a more shallow site. Combined with the ADCPs configured cell size and blanking distance the flow velocity at this location during low water levels could not always be determined accurately. The velocity profiles on the mid-channel tidal bar were recorded during two periods: the first period from 15 February to 12 April 2019, after which the batteries were renewed, and the second period from 24 April to 5 July 2019. All sensors positioned on the intertidal flat were upward looking Nortek Aquadopp Profiler 2 MHz sensors, mounted in the bed and equipped with an external battery pack to extend the measurement duration. Each instrument was configured with a cell size of 0.1 m and

a blanking distance of 0.1 m. Data were recorded during an averaging interval of 6 minutes and stored at an interval of 10 minutes.

During each field visit the height of each sensor head as well as the bed elevation of the transect was recorded with a Leica GPS1200+ system (Leica Geosystems, Switzerland) with Netherlands Positioning Service (NETPOS) as reference network. These transects were plotted to visualize small-scale morphological changes during the measurement campaign. Furthermore, three sediment samples of the top layer (5 cm) were taken at random locations around each sensor. These samples were mixed into a single container and frozen before transportation to the lab for analysis. For an overview of the intertidal bar morphology, we used LiDAR bed level data recorded at 5, 7, and 10 April 2019 and a multi-beam dataset for the area below -2 m NAP, recorded at April 18, 2019.

Wave data at the location WIEL, available every 30 minutes, and water level data at the locations VLIS, BRES and TERN, available every 10 minutes) were obtained from the measurement network of Rijkswaterstaat. Wind and pressure data at the location VLIS, available every 10 minutes, were obtained from the Royal Dutch Meteorological Institute. See Fig. 1(a) for the position of these locations and the section 'Data availability' for the web addresses.

# 140 2.3 Data filtering, reduction and analyses

135

Current velocity data were validated in four steps. First, the local water level at the sensor was used to remove all data from the cells that were not fully submerged. Second, all data from the sensor starting from one measurement cell below the surface and higher was also removed. Third, all data were checked for consistency in flow magnitude and flow direction by using a threshold value of 90 counts or higher for the average amplitude signal of the different beams of the sensor. Last, all remaining data were visually checked for irregularities and discontinuities, due to, e.g., turbulence or contamination. This data was manually removed. The local water level was obtained by interpolating the M2-tide parameters phase, amplitude, and mean sea level of the locations 'VLIS' and 'TERN' (Society, 1987).

To gain insight into the distribution of the current velocities during a full tidal cycle, the following steps were made. First, measurements near the bed, near the surface and averaged over depth were referenced to the time of local high water in classes of one minute, resulting in 721 classes. Second, classes with a number of samples less than a predefined threshold were removed. Third, within a moving window of 30 minutes, the 50% and 95% values of the current magnitude and the circular mean (Fisher, 1993) of the current direction were calculated. Through this data reduction all tidal cycles can be compared and statistics can be determined. The resulting series used for the analysis of flow velocity variation in the profiles for the 50% and 95% near-bed (NB), depth-averaged (DA) and near-surface (NS) velocities are referenced to as NB<sub>50</sub>, NB<sub>95</sub>, DA<sub>50</sub>, DA<sub>95</sub>, NS<sub>50</sub>, NS<sub>95</sub>. The circular mean of the direction data is referred to as direction D<sub>c</sub>. To discriminate between tidal and sub-tidal variations in water levels or flow velocities (only for data in channel), a Godin filter was applied to the time series (Foreman and Henry, 1989).

# 2.4 Sediment transport

To understand intertidal bar development, the gradients in sediment transport vectors are important. A proxy for the cumulative sediment transport at each location was calculated based on the near-bed and on the depth-averaged velocities. Sediment transport predictors for sandy suspended sediment typically have a dependence on the local flow velocity to the power three to five. We left out the critical shear stress in the proxy, since this does not alter the general pattern of sediment transport, but only amplifies the flood dominant character of the system. Apart from a scaling factor  $\alpha$  that depends on the sediment size, among other things, the heuristic behaviour of the sediment transport rate is  $Q_s = \alpha |U|^2 U \,\mathrm{m}^3 \mathrm{s}^{-3}$ . We calculated this quantity for each location (assuming  $\alpha = 1$ ) in along-shore (u) direction and cross-shore (v) direction and summed over i to obtain a proxy for the cumulative sediment transport as

$$Q_u = \frac{\sum_{i=1}^{N} \mathbf{u}_i \left( u_i^2 + v_i^2 \right) \right)}{N} \tag{1}$$

$$Q_v = \frac{\sum_{i=1}^{N} \mathbf{v}_i \left( u_i^2 + v_i^2 \right) \right)}{N}$$
 (2)

The total number of samples N of each location will vary, due to the fact that not all locations are fully submerged during a tidal cycle. Also, some locations were buried during some time due to sediment displacement. Sediment transport was calculated with the near-bed as well as the depth-averaged current velocity, in a rotated Cartesian reference frame: the u-direction is taken in the large-scale alongshore direction of the tidal flat, 115° with respect to the North (clockwise, positive towards the East). The cross-shore v-direction is oriented perpendicularly at 25° with respect to the North.

### 3 Results

#### 175 3.1 Hydrodynamic boundary conditions

Figure 3 displays the time series of the hydro-meteorological conditions. During the measurement campaign several neapspring tidal cycles with calm weather and storm conditions were captured. The water level at location 'VLIS' varied between a minimum of -2.73 m NAP and a maximum of 2.93 m NAP, the average wind speed was 6.2 ms<sup>-1</sup> coming from WSW direction. The first three weeks of March were characterized by stormy weather. During this period western wind speeds up to 25 ms<sup>-1</sup> (9 Beaufort) were registered, the peak significant wave height at the location 'WIEL' was up to 2.5 m. The setup of the water level during this period had peak values of 0.6 m at the field site. At 27 April and 7 and 8 June, wind speeds were over 13.5 ms<sup>-1</sup> (7 Beaufort), coming from a SW to W direction.

# 3.1.1 Large-scale flow patterns

In this paragraph we analyse the large-scale flow pattern. This pattern is calculated using the peak flow values during flood and ebb, averaged over the measurement period. Local differences are obtained from the depth-averaged time series  $DA_{50}$ ,  $DA_{95}$ 

and circular mean direction at each location. The local observations are analysed and used to construct the large-scale flow pattern.

The flow during average conditions, represented by the  $DA_{50}$  time series of each location is displayed in Fig. 4. In general, the magnitudes are largest in the channel and decrease higher up on the intertidal flat. The magnitudes at the locations of transect T1 on the intertidal flat are noticeably smaller than their counterparts on transects T2, T3 and T4. Peak flood flow is typically reached 50-60 minutes before high tide which is in agreement with the peak flow in the alongshore direction. The exceptions are locations '14' and '24'. At these two locations a clear peak during flood is missing: the flow is maximum at the moment the sensor is flooded and subsequently decreases. At both locations the cross-shore component has a considerable effect on the current magnitude. Slack water occurs for most locations up to 45 minutes after high tide. Exceptions are the locations on the mid-channel bar of transect T1 ('12, '13', '14') and the locations of transect T2 at higher altitude ('23', '24') where slack water occurs before high tide. The peak ebb flow occurs 1.5 to 2 hours after high tide. Noticeable are the differences between locations in the channel: the timing of the peak ebb flow at the locations '31' and '41' occurs more than 1 hour later and the magnitude is much larger than at the locations '11' and '12'. The peak ebb flow of locations higher up on the intertidal flat occurs earlier than those of the locations in the channel.

The flow direction is shown in Fig. 5. In the channel the flow is bidirectional and the transition from ebb to flood and vice versa occurs within 30 minutes. On the lower parts of the intertidal flat the flows are approximately bidirectional, the variance around the mean direction is more scattered. Higher up on the intertidal flat the flow direction is even more variable in time and at some locations a rotation is present in the transition from flood to ebb. On transect T1 the direction during flood rotates counter-clockwise towards the ebb direction and remains parallel to the local depth contour during ebb. On the intertidal flat at transect T2 the flow rotates clockwise, opposite to the direction of transect T1.




To analyse the effect of spring tides and events with large setup on the flow pattern, the  $DA_{95}$  series are compared to the  $DA_{50}$  series. In the channel the differences are minor: only the magnitudes are larger, but the timing of peak flood, peak ebb, and slack tide is similar. On the intertidal flat the differences are more pronounced: the  $DA_{95}$  magnitudes are much larger and the timing of peak flood, peak ebb and slack tide with respect to local high tide can differ: at location '14' the magnitude decreases towards the moment of slack tide, which is contrary to average conditions. At location '32' a pronounced maximum flow occurs three hours after high tide, which is ca. one hour later than during average conditions.

Figure 6 shows the large-scale velocity pattern at average ebb and average flood (for each location averaged over all measured peak ebb and flood flows). Note that at each location the length of the time-series is different, because of the sensor falling dry or missing data. The overall morphology-inferred pattern of ebb and flood dominant channels as shown in Fig. 1 is confirmed by the hydrodynamic observations: the majority of the locations are flood dominant as expected. Exceptions are location '31' which is clearly ebb dominant, and location '41' which has equal ebb and flood flow magnitudes. During flood the flow bends around the higher part at the westward side of the mid-channel bar. This causes relatively weak alongshore and strong cross-shore flood flows at the higher parts of transects T1 and T2. The peak flood flows in the channel are larger than on the mid-channel bar and for transects T3 and T4 the flows are mainly directed along local isobaths. At peak ebb the velocity direction changes approximately 180°, although there are some exceptions (locations '12' and '24'). For most locations the

near-bed, depth-averaged and near-surface velocities differ only a few degrees in direction, but at locations '11', '12' and '21' the difference in direction is more pronounced. At all locations the magnitudes increase from the bed to the surface.

# 3.2 Sediment transport






The sediment transport in the along-shore and cross-shore direction is first presented and the large-scale sediment transport pattern during ebb and flood is subsequently analysed. Our main focus is the transport of coarser material, therefore the near-bed and the depth-averaged sediment transport is addressed.

Figure 7 shows the cumulative along-shore and cross-shore sediment transport as a function of time at all locations based on the near-bed and depth-averaged velocities. At the channel locations '11' and '21' the net transport is in the direction of the flood current, the sediment transport onto the mid-channel bar at these locations is small. Locations '31' and '41' show opposite behaviour: the net transport is in the ebb direction, while location '31' has a component away from the mid-channel bar. The cumulative transport shows a steadily increasing or decreasing trend over time, suggesting that the transports are dominated by the tides. Locations '41' and '42', located in the flood-dominated channel terminus, show a strong modulation due to the neap-spring tidal cycle with reversing net direction of transport in calm conditions. Higher up the tidal flat the transport is much more influenced by episodic events. All locations on the tidal flat clearly show the impact of the large storm event in March 2019. During this event the cumulative alongshore transport shows a sudden change, after this event the behaviour is more gradual and for location '24' even in the opposite direction. The sediment transport is directed onshore for transect T2, but offshore in the upper parts of transect T1, suggesting a recirculation of sediment in line with the mean flows. At all transects the cumulative sediment transport proxy applied to the near-bed flow velocity is smaller than that applied to depth-averaged in alongshore as well as cross-shore direction. This is a direct consequence of the lower velocity in the near-bed cell compared to the depth-averaged velocity. While a lower magnitude is expected, our interest lies in the change in direction of near-bed flow due to three-dimensionality of the flow around the bar.

The large-scale sediment transport pattern for mean flow during ebb and flood is shown in Fig. 8. The NB<sub>50</sub> and the DA<sub>50</sub> shown in this figure are used to calculate the sediment transport according to equations 1-2. During flood, net sediment is transported onto the mid-channel bar at transects T1 and T2. In the eastern part of the field site, i.e. transects T3 and T4, the net sediment transport in the flood direction mainly takes place along the isobaths, with the exception of location '31' where sediment is transported into the ebb direction. The results show that differences in magnitude as well as direction between depth-averaged and near-bed transport are present, although differences in direction are small ( $< 2^{\circ}$ ).

#### 3.3 Development of the bed surface elevation in the cross-shore transects

Each month, the field site was visited and the height of the bed along the transects was recorded, except in March 2019 due to severe weather conditions. The recorded bed levels are shown in Fig. 9. At transect T1 we observed sedimentation during the period February to April. Nearby location '13' the sedimentation amounted to 0.57 m and towards location '14' to 0.25 m. These locations also had the strongest veering. After this period the sediment deposited did not erode, the area between the two locations was filled with sediment resulting in an accretion of this area of 0.35 m during the measurement period. Transect

T2 accreted gradually but mildly between February and July with at most 0.17 m along the transect. During the measurement period, the bed level on the tidal flat gained maximally 0.10 m in height at transect T3 and T4. In the profile below -1 m NAP at transect T3 the ridge that was present in February eroded during the measurement period. In Fig. 1(c) the elevation over a period of 10 years is shown. Comparing this with the elevation measured during the field campaign, we see that an above average accretion has occurred at transect T1 and to a lesser extent at transect T2. The height development along transects T3 and T4 is in agreement with the 10 year average.

### 260 4 Discussion



First we discuss the horizontal patterns of cross-shore currents and contrast these with the potentially simpler shore-connected tidal flats as conceptualised in a commonly used mathematical model for cross-shore flow. Second, we discuss the horizontal patterns of along-shore currents and contrast these with a mass balance for along-shore flows for areas defined by our measurement stations. Third, the vertical profiles of flow velocity are compared between the stations to assess the potential effect of veering on sediment transport. Finally, we discuss how morphology and external forcing affect the large-scale sediment transport patterns.

### 4.1 What drives the cross-shore flow?

The cross-shore flow velocities are important for sediment transport onto and off the tidal flat. Since we know that the midchannel bar has a 3D morphology and is not alongshore uniform, we are interested in whether a 1D model is suitable for such a location. Therefore, to further understand what drives the cross-shore current, we compare observed depth-averaged cross-shore currents with those obtained by the 1D model as described in Friedrichs and Aubrey (1996). In this model, it is assumed that water levels are spatially uniform over the tidal flat and that morphology of the tidal flat is alongshore uniform. The depth-averaged cross-shore tidal current  $U_T(x,t)$  can then be described by:

$$U_T(x,t) = \frac{x_f(t) - x}{h(x,t)} \frac{\mathrm{d}\eta(t)}{\mathrm{d}t}$$
(3)

where

 $x_f(t)$  = the boundary between the wetted and exposed portions of the tidal bar

x = the cross-shore position on the profile

t =the time

h(x,t) = the water depth, which is the difference between the water level and the bed level

$\eta(t)$  = water level, assumed to be spatially uniform.

We applied this model to each transect to calculate the resulting cross-shore velocities on the intertidal flat. The bed profiles were taken from the LiDAR dataset recorded in April 2019 and smoothed with an averaging window of 100 meters. The water

Table 1. Coefficients of linear regression between the measured cross-shore velocities and a 1D numerical model

| Transect | Location | $R^2$ | a    | b      |
|----------|----------|-------|------|--------|
| 1        | 2        | 0.68  | 0.21 | -0.084 |
|          | 3        | 0.13  | 0.17 | -0.013 |
|          | 4        | 0.08  | 0.34 | -0.024 |
| 2        | 2        | 0.74  | 0.36 | 0.008  |
|          | 3        | 0.87  | 0.58 | 0.005  |
|          | 4        | 0.95  | 0.69 | -0.013 |
| 3        | 2        | 0.40  | 0.44 | 0.005  |
|          | 3        | 0.69  | 0.61 | -0.002 |
|          | 4        | 0.66  | 1.00 | -0.017 |
| 4        | 2        | 0.21  | 0.40 | -0.002 |
|          | 3        | 0.81  | 0.69 | -0.010 |
|          | 4        | 0.91  | 0.45 | 0.003  |

levels for each transect were based on the time series of channel locations ('11' to '41'; see methods section). Since the 1D model does not account for full submersion of a transect, which did occur during several tidal periods, the maximum value that  $x_f$  could obtain was set to the distance of the highest point measured from x = 0, i.e., the channel location. The measured depth-averaged velocity was split into a cross-shore and along-shore direction. The local along-shore direction was taken parallel to the local isobath derived from the smoothed LiDAR bathymetry, the cross-shore direction was taken perpendicular to the along-shore direction, positive towards the tidal flat according to Friedrichs and Aubrey (1996).

The correspondence between modelled and measured cross-shore velocities was quantified by linear regression, the coefficients are shown in Table 1. Most locations have  $R^2 > 0.6$ , whereas the locations on transect T1 located behind the elevated area, i.e., locations '13', and '14' and the locations near the channel of transects three and four, i.e., location '32', and '42' have  $R^2 < 0.50$ . Although at many locations the correlation coefficient  $R^2 > 0.65$ , the slope (a) is in general much smaller than one, typically around 0.5, which means that the measured cross-shore velocities are about twice as large as the modelled ones. For transect T1 and T2 this is consistent with the observation that the flow rotates onto the tidal flat, thereby adding momentum to the cross-shore component with higher velocities as a result. It is known from field visits during this campaign that the bed-level is not uniform and differs in composition spatially and temporally. We looked into the relation between flooding of the intertidal bar and the effect on the flow, but no relation was found.

# 4.2 What drives the along-shore flow?



To gain better understanding of the role of along-shore flow in filling and emptying of the intertidal flat, we solved the mass balance for all quadrangular areas between the measurement positions at the field site. These areas are depicted in Fig. 2 with

dashed lines and numbered one to nine. The mass balance of an area 'A' is calculated based on the definitions shown in Fig. 10 (a). The discharge Q through each boundary with length L is calculated as:

$$Q_1 = \frac{h_{11}\mathbf{u}_{11}\hat{n}_1 + h_{21}\mathbf{u}_{21}\hat{n}_1}{2}\Delta L_1 \tag{4}$$

$$Q_2 = \frac{h_{11}\mathbf{v}_{11}\hat{n}_2 + h_{12}\mathbf{v}_{12}\hat{n}_2}{2}\Delta L_2$$
 (5)

$$Q_3 = \frac{h_{12}\mathbf{u}_{12}\hat{n}_3 + h_{22}\mathbf{u}_{22}\hat{n}_3}{2}\Delta L_3 \tag{6}$$

$$Q_4 = \frac{h_{22}\mathbf{v}_{22}\hat{n}_4 + h_{21}\mathbf{v}_{21}\hat{n}_4}{2}\Delta L_4 \tag{7}$$

The mass balance equation for a single area reduces to:


$$\sum_{i=1}^{4} Q_i = -A \frac{\partial \eta}{\partial t} \tag{8}$$

where A is the surface of the area. Note that the area in practice is not square since the measurement locations are not placed at an equal distance. Figure 10(b) shows the results for the area between the locations '22', '32', '23', and '33': the sum of the discharge calculated over the boundaries is in agreement with the discharge derived from the water level variation times the surface of the area.

We analysed the along-shore and cross-shore discharge to understand their contribution to the total discharge of the area. Figures 11(a), (b) show the times series. During flood the net flow in the along-shore direction is positive (eastern direction), as a result of which the area is emptied. In the cross-shore direction the net flow is negative (southern direction), filling the area. Since during flood the net influx must result in an incremental water level, the net cross-shore discharge must compensate for the loss in along-shore direction resulting in a large cross-shore discharge at the boundary 'Q2'. During ebb the net flow in the along-shore direction is negative, i.e., the area is filled from the east, while the net flow in the cross-shore direction is positive (outflow to the north) and thus empties the area. Based on the flux quantities we conclude that part of the along-shore discharge at 'Q3' is added to the cross-shore discharge at 'Q2'. The net discharge created by the along-shore flow is compensated by the cross-shore discharge resulting in an increase of the cross-shore velocity. This explains the larger cross-shore flow velocities measured during flood and ebb than those based on the 1D approach presented earlier where the along-shore flow did not contribute to the discharge.

Although model results for this area are satisfactory, the model fails at the other areas. A possible explanation is that the areas defined by the measurement locations are rather large compared to the observed rotational patterns as shown in Fig. 6, so that we lack information on flow behaviour in between the measurement locations.

# 4.3 Large-scale sediment transport patterns on mid-channel bars and their dependence on hydrodynamics and morphology

In this section the effects of hydro-meteorological conditions and morphology of a mid-channel bar on large-scale sediment transport patterns are described. First, we consider the effect of wind speed on the current flow in relation to the sediment transport. Second, we relate local morphology to large-scale sediment transport patterns.

The results of the proxy for sediment transport showed that the cumulative transport at locations '11', '21', and '31' is a steady process, while at location '41' and at the mid-channel bar it has a more episodic character. For the locations on the mid-channel bar a probable cause may be the stronger influence of wind on the shallower parts of the tidal flat (Hir et al., 2000; de Vet et al., 2018; Colosimo et al., 2020). To quantify the effect of wind on the peak velocities we compared the peak current velocities during situations with almost no wind (wind speed less than 3 Beaufort) with those occurring during periods with high wind speed (more than 6 Beaufort). To do so, we proceeded with the following steps. First, we divided the peak velocity data into two subsets: a weak-wind selection containing peak velocities during wind speeds smaller than 3.3 ms<sup>-1</sup> (3 Beaufort), and strong-wind selection containing peak velocities with wind speeds larger than 10.8 ms<sup>-1</sup> (6 Beaufort). For both datasets the tidal range associated with the peak velocity was also registered. Second, for each measurement location a linear fit was made between the tidal range (X) and the peak velocities for ebb as well as flood (Y). Third, the obtained regression coefficients were used to calculate the peak velocities for strong wind given the tidal range in the dataset. Finally, to determine the effect of higher wind speed, the fraction between measured and calculated peak velocities was calculated. These are shown for flood and ebb on a logarithmic scale in Fig. 12. We see the most pronounced effect on the current flow at the shallowest parts of the tidal flats, especially during flood. This strong increase of flow velocities throughout flood high up the tidal flat also explains the more episodic character of the proxy for sediment transport at the highest parts of the tidal flat. During ebb when wind and tide are in the opposite direction, there is a reduction of the peak flow, although at the most shallow locations the current flow is amplified, despite the ebb current. The mean tidal range for both periods is almost the same: 3.72 m for weak wind and 3.68 m for strong wind. Therefore, changes in the peak velocities will be mainly caused by the wind stress.

The flood- and ebb-dominant channels influence the ebb and flood dominance on the mid-channel bar: while locations '11' and '21' are clearly flood dominant, there is a sudden spatial transition to ebb dominance for location '31'. This could be caused by flooding of the northern sand spit during ebb from the adjacent ebb-dominant channel, thus affecting the flow direction at the locations '31' and '41'. Although this cannot be confirmed due to a lack of measurements we hypothesise that multiple channels with a different flow regime will affect the net flow on a mid-channel bar locally. This subject has not been studied previously. On the other hand, a change in global flow patterns due to the tidal gradient between adjacent channels during flooding of the mid-channel bar as suggested by de Vet et al. (2018) was not found. Local morphology introduces veering of the flood current and is therefore potentially important to the sediment transport onto the mid-channel bar. We hypothesise that these patterns, initiated by local morphology, will also occur on the edges of fringing tidal flats if these are constrained by groynes or transverse dikes. Seaward stretching dams also likely impact the along-shore current and thus influence the sediment transport (Wang et al., 2019).

# 5 Conclusions







In this paper we analysed current velocities measured on a mid-channel bar at 16 locations during a period of six months to gain insight into large-scale flow patterns. To understand how a mid-channel bar is filled and emptied, we compared the velocities

with the result of a 1D model for fringing tidal flats. The effect of hydrometeorological conditions and local morphology was obtained by analysing the large-scale flow and sediment transport patterns. This led to the following conclusions:

In contrast with open and fringing tidal flats, the cross-shore current is not the sole responsible driver for filling and emptying a tidal flat on a mid-channel bar. Instead, the along-shore current strongly affects the large-scale flow patterns. This is clearest in the veering at the head of the mid-channel bar, thereby adding momentum to the cross-shore current. As the finite length of a mid-channel bar implies that the along-shore flow cannot be uniform, we deduced from the data that the along-shore flow is also driven by gradients in water level caused by local morphology. This result points to the effect of the bar shape on the flow pattern, which invalidates the common assumption that tidal flats are spatially uniform and only cross-shore current needs to be considered. Whether the assumption of non-uniformity of the along-shore flow along fringing tidal flats holds when dams or groynes are present requires further research.







We used a proxy to estimate the sediment transport. Based on this proxy we observed differences in direction and magnitude between values nearest to the bed, i.e., the lowest measurement cell, and the depth-averaged value. Although magnitude differences in the vertical profile can be parametrized in 2DH models, the question remains whether and how differences in direction over the vertical profile can be solved using such models, or that more sophisticated models are needed.

The commonly used cross-shore flow model valid for uniform tidal flats, and a model for along-shore flow calculated from the mass balance, are of limited value because these models neglect rotation of the flow between the measurement points. As local morphology and tidal conditions determine these flow patterns, the question is how representative local measurement points can be for specific areas. This insight is important when validating numerical models with local measurements: the placement and spacing of instruments should fit the spatial structure of the field site. If this area is alongshore uniform, a cross-shore line-up is sufficient. When the main spatial changes are dominated by the alongshore velocity component, a line-up in the alongshore direction is needed. When the study area has heterogeneity in all directions, the instruments may be placed in a cross line-up, or as an array as used in this study.

During events that amplify flow, i.e., storm conditions with high wind velocities aligned with the net tidal current direction, sediment transport is also amplified. In coastal regions and estuaries such as the Western Scheldt where sediment is relocated by dredging and disposal, knowledge of sediment pathways is important to preserve or maintain intertidal habitat area. For further studies that take morphological changes and sediment transport into account it would be useful to identify possible sediment sources, including locations of disposal sites and quantities of disposed sediment. Despite the tidal dominance of our environment, it would be valuable for insight and model validation to look into possible effects of waves on the large-scale sediment transport patterns in the intertidal areas, numerically as well as through measurements of flow and sediment transport in suspension and on the bed.

Based on our results, we conclude that velocity patterns and concomitant sediment transport on a mid-channel bar are more complicated than assumed in often used along-shore and cross-shore models. Our analyses of hydrodynamics and morphodynamics show that specific morphological features associated with mid-channel bars result in deviations from commonly assumed flow patterns on tidal flats. Elucidating these processes and assessing how important the intricate patterns are for

sediment transport requires a combination of numerical modelling and measurements that cover the spatial variations of the study area.

Data availability. Datasets of current velocity, bed elevation profiles and water levels of the measurement locations are available at Zenodo (https://doi.org/10.5281/zenodo.15017660). Wave data and water level data are available at the website of Rijkswaterstaat (https://waterinfo.rws.nl). LiDAR data is available at Rijkswaterstaat (https://rijkswaterstaatdata.nl). Wind en pressure data is available at the website of the Royal Dutch Meteorological Institute (https://www.knmi.nl).

*Author contributions.* MS: data analysis, code development, writing (original draft, review and editing). MvdV: data analysis and writing (review and editing). GR: review. MK: supervision, review and validation.

Competing interests. The authors declare that they have no known competing financial interest or personal relationship that could have appeared to influence the work reported in this paper.

Acknowledgements. This work is part of the PhD project of Marco Schrijver which is funded by Rijkswaterstaat Zee en Delta. We would like to thank an anonymous reviewer and reviewer Bram van Prooijen for their comments that helped to improve the paper. We are grateful for the help from Rosanna van Hespen (https://10000words.nl/) and Mascha Dedert during the writing process. We specially thank Jan de Bel, Mariska van Dam, Geert den Hartog (dec. 08-03-2020), Arno Slager and Jan van 't Westende (dec. 28-05-2023) of Rijkswaterstaat CIV for their contributions to the measurement campaign.

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

**Figure 1.** (a) Overview of the western part of the Western Scheldt in the Netherlands. The ebb and flood patterns are derived from model calculations (ScalWest) with a representative spring tide as boundary condition and the bathymetry of the year in question. (b) Height of the mid-channel bar 'Hooge Platen'. (c) Sedimentation/erosion between 2009 and 2019 in meters. Height is relative to Dutch Ordnance Level (NAP).

**Figure 2.** Overview of field site located at the mid-channel intertidal bar 'Hooge Platen' in the Western Scheldt. Four transect (T1 to T4) with instruments are shown. Instruments are labelled by their transect number (first position) and their position on the transect (second number). For areas 1 to 9 the mass balance is solved based on the measurements.

**Figure 3.** Time series of the hydro-meteorological conditions. (a) high and low water level at location '11'; water level setup at (b) external (WIEL, blue) and local (11, red); (c) wind speed (VLIS), wind speeds of 3 and 8 Bft are displayed as red dashed lines; (d) wind direction (VLIS); (e) Significant wave height (WIEL). (f) Wave period (WIEL). The lunar phases are shown at the top. The locations are shown in Fig. 1(a) and Fig. 2.

**Figure 4.** Depth-averaged current magnitude relative to time of high water for each location with 50th percentile (magenta), and the 95th percentile (red). Time of high water is indicated with the dashed line.

**Figure 5.** Depth-averaged current direction relative to time of high water for each location with the circular mean (magenta). Time of high water is indicated with the dashed line.

Figure 6. Average peak flood and peak ebb velocity near the bed, depth-averaged and near the surface.

**Figure 7.** Cumulative sediment transport, near-bed and depth-averaged, in along-shore direction (positive towards east) and cross-shore direction (positive towards north).

Figure 8. Sediment transport during ebb and flood for average peak flow near-bed and depth-averaged.

Figure 9. Height along the transects measured during the field campaign.

Figure 10. Mass balance of area 5 (see Fig. 2 for locations). Panel (a) shows the methodology. Panel (b) displays the total discharge.

**Figure 11.** Discharge of area 5 in (a) along-shore direction, and (b) cross-shore direction.

Figure 12. Effect of wind on the current peak velocity for flood and ebb calculated as  $\log_{10}\left(\frac{U_m}{U_c}\right)$ , with  $U_m$  the measured peak velocities at wind speeds above 8 Beaufort, and  $U_c$  the peak velocities calculated with the use of the coefficients derived from situations with wind speeds less than 3 Beaufort.