# Peer review of "Hydrodynamic and morphodynamic patterns on a mid-channel intertidal bar in an estuary"

_EGUsphere, 2025_

## Referee Comment (RC2)

Dear authors,

I have read your manuscript with great interest. There is no doubt that this unique data set needs to be published (16! ADCPs over a period of 6 months, including a storm season). I am however not sure whether the chosen format is the best. You might consider changing the manuscript to a hydrodynamics-only paper. Leaving out the somewhat artificial way of dealing with sediment transport (a power of the velocity and neglecting waves) gives some more room for detailed analyses of the hydrodynamics. As indicated below, my main suggestions are related to the discussion part (the mass balances for cross shore and longshore). I hope these suggestions help in improving the paper.

Best regards,

Bram van Prooijen

Introduction

Line 44-54: The flow and sediment transport on mid-channel bars is described. It is suggested in L53-54 that the flow over the flats (differences with fringing flats) will likely have an effect. This has been studied already in De Vet et al (2018), where such flows (especially during wind events) occur and where it is shown that the combination of wind and waves are crucial for sediment transport. See also the work by Elmilady et al. (e.g. 2021).

Line 55: this line needs some nuance, as there is quite some quantification by numerical modelling, but field measurements are indeed not so abundant.

Overall, the literature review could be a bit broader, like including the work of the groups from San Pablo Bay (Lacey), the Seine and other French estuaries (Grasso, Verney, le Hir), the German systems, Chinese systems like the Yangtze Estuary or Jiangsu coast, the Danish Wadden Sea? Not all need to be included, but some would embed the paper better in literature.

Methods

Field site: this section contains a lot of Dutch names that are not further used in the Results or Discussion section. You could simplify it and refer to various other papers which elaborate on the history of the Western Scheldt and tidal flats. For example references would be needed to back up the last sentence "This rapid elevation decreases the intertidal area and increases the percentage of silt in the bed.".

Data Gathering: This is the strong point of the paper. The set up with so many ADCPs is unique. This might be emphasized a bit more here, but also in the abstract.

Data Filtering: L131-132: It is not fully clear for me why the data at VLIS and TERN were needed, as the ADCPs also record water levels. L133: what does manually checked mean, what was done, what were criteria?

Sediment Transport: The approach to determine sediment transport is fully based on the flow; waves are not considered. This should at least be mentioned here and later on in the discussion. Waves will likely have a substantial effect on sediment transport. Why is chosen for a formulation without a critical shear stress?

Results

Large scale flow patterns: L175-185 The peak flow around 50-60 min before high tide is likely a peak in longshore velocity. The peaks at 14 and 24 will likely have a cross-shore component too. This could be more emphasized here or in the discussion on the flow bending around the higher area in the west. The description in this part is somewhat unsatisfying, as it is not fully clear why these phase shift are given. Are they to be discussed in the discussion? L192-197: The spring tide ebb peak in 31 and 41 is (much) more pronounced and this peak also seems to be present in 11, 12 and 32 for spring tide. This might be related to a dewatering process that is stronger during spring tides. Most spring tide flood peaks seems to be a bit later than the normal tide flood peaks.

Sediment transport: L218 indicates a modulation by the spring-neap tidal cycle. This might be phrased stronger: sediment transport occurs during spring tide and almost nothing happens during neap tides. Only at 12, the alongshore component shows some changes during neap. The effect of the episodic event is indicated. This effect will in reality be much more important as waves will play a role in eroding the bed. This is relevant to note here.

L223: "At all transects the cumulative sediment transport near the bed is smaller than the depth-averaged transport in along-shore as well as cross-shore direction." This is a direct consequence of the smaller velocity near bed than depth-averaged. You could consider to

plot only the transport based on the near bed velocity, as you consider bed load. Having said that: the velocities are quite high, so suspended load might be important here. Did you check Rouse numbers?

Section 4.1

A 1D model is proposed, based on the rigid-lid approach and the subsequent volume balance. Such a model assumes that the width is uniform and that the flow is purely in one direction. A hard boundary condition was imposed at the highest point in the profile, assuming that flow cannot flow over the highest point (i.e. there is no net flow over the bar.

To compare the model results with the measurements, a linear regression is proposed (L270). It confuses me a bit why such an approach is chosen, as the model is based on a volume balance, there is no calibration parameter or so. What does the values for a and b thereby mean? You could consider to plot the results and indicate that they are all below the 1:1 line. This suggests that (i) the assumed control volume (no uniform width); (ii) or the boundary condition (no flow at the highest point); or (iii) the assumption of no longshore gradients; or (iv) the assumption of a uniform water level (no friction) does not hold (a better definition of these four might be needed). By only presenting the values for a and b, it is not clear during which parts of the tide the model is performing well and when not (is there a difference between neap or spring, or between ebb or flood?). Flow over the highest point would only be of importance during emergence of that highest point. The potential causes (i-iv) could/should be unravelled in more detail. Based on the present results, it is not clear which cause is most important. It is therefore also unclear what the cross shore current drives.

For example, the assumptions of a 1D model are quite violated for Transect 1. There is net cross shore flow velocity and likely also discharge? (Figure 6); there is likely a longshore gradient in longshore velocity; friction will likely play a role during dewatering. The values for a for Transect 1 are therefore much smaller than 1.

This section needs to be reconsidered.

Small notes:

L263: why was smoothing with a 100m window needed?

L276: the model does not need a bed level with a constant slope (linearly changing bed level). The model can cope with a convex-up slope.

L276: contraction -> contradiction?

Line 279: the model assumes that the longshore flow is uniform, not zero, see also L278.

Section 4.2

To explore the causes for the longshore current, another volume balance is considered. In this case, the balance is fully based on measured velocities and depths. If the aim is to explore the causes for the longshore velocities, I would rather use a momentum balance. In this way, the contributions of pressure gradients, bed friction, wind stress, advection, etc could be quantified. The volume balance is now only presented for a single area and only for two tidal cycles. In L305 it is even mentioned that the method does not work for the other areas. It would be valuable to explore whether there are differences in dominating terms between spring tides and neap tides. I would also expect that wind events would have a different balance between the different terms. Furthermore, it would be interesting to compare the difference between lower and higher area for the balance between the various terms: is friction more important on the higher parts?

In its present form (volume balance), it is unclear what drives the along-shore flow. A reconsideration would be needed for this section too.

Section 4.3

The title is "*Large-scale sediment transport patterns on mid-channel bars and their dependence on external forcing and morphology*". It suggests a dependence on external forcing. The sediment transport is however based on the local velocity, which is not an external forcing. Two aspects are considered (L310-313): effects of wind and effects of local morphology. An aspect that I miss is the effect of the neap-spring tidal cycle. It is mentioned a bit in the Results section. In the analysis on the effect of the wind, it is not clear whether the effect is caused by the surge or by the local wind stress. As indicated earlier, the waves will have a crucial role here too.

L316-320: the data set is split into two parts, based on wind speed. It is not clear which periods are in which category, nor why the Bf3 and Bf6 are chosen. Are there periods with wind >6Bf but from another direction than from the west? It would be helpful to indicate the bands of mild conditions and storm conditions in Figure 7. It would then become clearer whether wind affects the flow (sediment transport).

Figure 12 and associated text is very difficult to follow and interpret.

The ebb dominance of the channel between the main bar and the spit could get some more elaboration. The explanation in L330 is a bit short. Can you explain more about the dewatering of the shoal via that channel?

Figure 2 shows a clear difference in bedform patterns. Are these differences to be considered too?

Conclusions

343-350: The assumption is made that it is commonly assumed that cross-shore velocities are the only driver for filling/emptying a tidal flat. This is maybe somewhat overstated. Other authors (e.g. de Vet et al) also indicated already the more complex flow patterns on shoals.

L344: It is not convincingly shown that 2DH models overestimate bed load transport. The argument given in the next sentence mention a representative time period. These seem two different aspects: (i) is a 2DH model able to represent the flow?; (ii) can a representative period be used for upscaling? A statement about the validity of numerical models would need more elaboration.

The impact of waves is not considered in this study. It should at least be mentioned somewhere. How would the wave impact change the conclusions?

L370: this is a bit demotivating for almost all readers, especially as last sentence. Even 16 ADCPs are not enough to identify the main flow patterns? Most readers will likely have much less instruments. It would be valuable to indicate what would be possible with less instruments. Would you place them in a transect or in a cross?

---

## Author Comment (AC1)

General

The paper is well written and is almost ready for publication.

Nevertheless the authors conclude that even this intensive measurement campaign is to short and the resolution is too low to fully understand the complex flow patterns and sediment transport, still he dataset is valuable for publication.

There are a few general remarks I would advise to elaborate or to comment on in more details to improve the manuscripts:

- In the introduction the authors state appropriately that one of the main forcings for the sediment transport at intertidal flats is the (ship induced) wave exposure. Wave exposure was not measured or modelled for the field side. Nevertheless, during the six month measurement campaign a 3 week period of stormy condition occurred. The field side is close to the estuary mouth area and the main shipping lane towards the seaports of the Scheldt Estuary. Therefore, the authors should elaborate a bit more on the expected wave impacts at the field side and their consideration for not measuring, or modelling wave exposure at the field side.

  Reply: We agree that waves may be important in relation to the sediment transport patterns, if high magnitude waves occur often enough to modify the current-driven morphodynamics. However, we don't have enough spatial information to quantify the effect of waves in detail and we therefore decided to focus in this manuscript solely on the role of currents. To gain better insight into the effect of waves especially during storm conditions and higher up the intertidal bar, the effect of waves will be addressed in a follow up study using a high-resolution numerical model that will also include the effect of waves. The effect of waves induced by vessels is, due to the distance of the navigations channel and the shallow bar 'Spijkerplaat' located between the channel and the field site , is assumed to be very small.

  We suggest the following change to the Introduction (L64-69)-:

  For this, we analysed the measured the current velocities on a spatial grid at 16 locations for a period of 6 months on a mid-channel tidal bar in the Western Scheldt estuary in the Netherlands. The results are compared to simple models developed for open and fringing flats, which serve as a starting point for understanding differences with mid-channel bars. The measured cross-shore and alongshore velocities were used as input into a simple sediment transport predictor to understand how the hydrodynamic forcing and tidal flat morphology determine the sediment balance of a mid-channel tidal bar. The effect of waves on the sediment transport, as described in among others (Roberts 1999, LeHir 2000, Green 2013, Hunt 2015, Maan 2015, Hunt 2016, de Vet 2108) was omitted in this study since

spatial information to quantify the effect of waves on sediment transport was missing.

- A very simple sediment transport approximation is used (alpha x u^3). To my opinion, this could be extended to some extra transport formula like Engelund-Hansen or Van Rijn formula which are commonly used in numerical models. Rather than comparing bed load to depth averaged transport in de simplified approach, which is in my opinion incorrect (see comments below), I would advise to compare the different models for total load. Transport calculations are only based on velocity measurements. A sediment concentration measurement (maybe for a short time and limited number of locations) would have be an added value to validate the approach. The authors can comment on that in the conclusions.

  Reply: Our interest is to gain insight into the large scale patterns and the forcings that steer the sediment transport and the spatial gradients thereof, which drive morphological change. Therefore a simple proxy was used, based on the measured current velocity. The use of different predictors will have no great impact on the qualitative patterns found, since these also rely on the current velocity in a nonlinear way. An Engelund-Hansen predictor will have a dependency on flow velocity to the power five while we used a power of three. This will change magnitudes, not patterns. Sediment concentrations were not measured during the campaign. In the conclusion section (L365-366) it is recommended to attempt measuring suspended sediment transport in future campaigns.

- The authors highlight and show the importance of wind on the sediment transport by comparing a period with strong wind conditions to a calm period. However, from the manuscript it is not clear which is the most important driving mechanism: is it the local wind shear stress causing extra wind induced currents in the channel and shallow intertidal area, or is it a more global effect of SW to NW winds and atmospheric pressure fields associated with these that cause extra surge during rough and stormy conditions leading to higher high waters and/or tidal range and thus longer submerging time of the tidal flats? From a modelling perspective, it would be very valuable to go more in depth on this, merely since the authors highlight the sensitivity of 2DH models to external forcings.

  Reply: see our reply on remark Line 320-325

Minor remarks on the text:

- Line 26: For sediment transport patters not only direction of the tidal flow and wave exposure are important, also strength of the tidal flow, tidal range, grain sizes and sand-silt ratio.

Reply: This will be added to the text in the revised manuscript.

- Line 56: For many estuaries 3D models are developed as well. When only investigating currents and sediment transport, computational cost is not that much an issue anymore. Unless for long term morphological simulations. However, often these 3D model tend to perform poorly in very shallow areas like on tidal flats. Another technical issue is wetting and drying which still remains a technical challenge from a numerical point of view.

  Reply: We agree on the fact that 3D models are available for different estuaries and that there are still issues on their performance.

- Line 87: The Western part of the Scheldt Estuary is very dynamic, specially the Schaar van Spijkerplaat. How stable is the channel where the measurement (points 1) took place?

  Reply: The channel was not stable during the measurement period: sedimentation occurred as is shown in Figure 1 of this reply. As is described in Lines 111-112 of the paper, the sedimentation did affect the measurements at location MP4.

[Figure]

*Figure 1 Difference in meter between the surveys from 24-06-2019 and 28-02-2019. Red is sedimentation, blue is erosion.*

- Line 99: How is rapid bed elevation increasing the percentage of silt in the bed

  Reply: We suggest the following change to the text:

  L99: Peak flow velocities decrease as the bed level becomes higher. As a result, finer sediments will settle easier, thereby increasing the percentage of silt in these areas.

- Line 139: what was the threshold

  Reply: We suggest the following change to the text (L139-140):

  Second, classes with a number of samples less than a predefined threshold were removed. This threshold was visually determined for each location, since this value depends on the number of representative samples per timestep in the dataset. Third, within a moving window of 30 minutes, the 50\% and 95\% values of the current magnitude and the circular mean (Fisher1993) of the current direction were calculated.

- Line 152: in the equation: ms$^{-1}$ is the unit, place between brackets.

  Reply: All units in the text are without brackets as instructed in the journal format.

- Line 167: During the stormy periods, what wave heights are estimated at the measurement site? Was it measured, a wave model could give some insights?

  Reply: see our reply to the general remarks.

- Line 172: Why are the peak ebb and flood currents averaged over the full measurement period? Peak currents can be significantly different between neap and springtide.

  Reply: Our interest is the general sediment transport pattern and the differences in the vertical profile. Therefore the averaged peak flood and ebb velocities over the full measurement periods of the lowest cell, the top cell, and the calculated depth averaged are used.

- Line 183: Ebb currents at 31 and 41 differ indeed from what one would expect like in 11 and 21. Is there an explanation for? The gully looks like a floodgully.

Reply: the ebb dominant character of these two locations could possibly be explained by dewatering of the shoal and the flow coming from the ebb dominant part of the channel 'Schaar van Spijkerplaat'. The last one contributes to the current flow at these locations as long as the northern sand spit is submerged.

- Line 225: Since the transport is only calculated based on the measured currents without taking the concentrations into account, this conclusion is in my opinion presumptuous. Generally the formulation $u^3$ holds only for the total load with u the depth averaged velocity. I don't think it makes sense to differentiate between total load and bed load.

  Reply: we use the proxy to gain insight into the possible effect of subtle differences in the velocity profile on sediment transport magnitude and direction. We are aware that we need the vertical sediment concentration profile as well to quantify the full effect of the change in flow velocity direction, but we don't have these available. However, we are convinced that our approach shows the same effects. Typically, sediment concentrations are much larger near the bed than higher up in the water column. The sediment transport direction is therefore largely influenced by the direction of the near-bed flow velocities. This is the reason why we chose to compare sediment transport directions between a $u^3$ calculated with depth-averaged and near-bed velocities. Hence, this proxy offers a quick scan of the impact of the current velocity at different depths to the sediment transport direction.

- Line 320-325: Needs some more clarification and motivation on why and how DS2 is compared to the linear regression of DS1. At least for some of the station: make a scatterplot with the linear regression and DS2 peak velocities. From the text it is not clear how the wind is influencing the peak velocities: is it a direct effect of the wind shear stress. Or is it due to the fact that SW tot NW winds cause a higher surge which lead to higher tidal amplitude?

  Reply: We suggest the following change to the text:

  L316-324: To quantify the effect of wind on the peak velocities we compared the peak current velocities during situations with almost no wind (wind speed less than 3 Beaufort) with those occurring during periods with high wind speed (more than 6 Beaufort). To do so, we proceeded with the following steps. First, we divided the dataset containing peak velocities into two subsets: dataset one (DS1) containing peak velocities during wind speeds smaller than 3.3 ms$^{-1}$ (3 Beaufort), and dataset two (DS2) containing peak velocities with wind speeds larger than 10.8 ms$^{-1}$ (6

Beaufort). For both datasets the tidal range associated with the peak velocity was also registered. Second, for each measurement location a linear fit was made between the tidal range (X) and the peak velocities for ebb as well as flood (Y). Third, the obtained regression coefficients were used to calculate the peak velocities of DS2 given the tidal range in the dataset. Finally, to determine the effect of higher wind speed represented in DS2, the fraction between measured and calculated peak velocities was calculated. These are shown for flood and ebb with a logarithmic scale in Figure 12. We see the most pronounced effect on the current flow at the shallowest parts of the tidal flats, especially during flood. This strong increase of flow velocities throughout flood high up the tidal flat also explains the more episodic character of the proxy for sediment transport at the highest parts of the tidal flat. During ebb when wind and tide are in the opposite direction, there is a reduction of the peak flow, although at the most shallow locations the current flow is amplified, despite the ebb current. The mean tidal range for both periods is almost the same: 3.72 m and 3.68 m for DS1, resp. DS2. Therefore changes in the peak velocities will be mainly caused by the wind stress.

To clarify Figure 12 we propose to change the caption to:

The effect of wind on the current peak velocity for flood and ebb calculated as $\log_{10}\left(\frac{U\_\{measured\}}{U\_\{calculated\}}\right)$, with $u_{measured}$ are the measured peak velocities at wind speeds above 8 Bft, and $u_{modelled}$ are the peak velocities calculated with the coefficients derived from the situations with wind speeds smaller than 3 Bft.

- Line 350: see comment before on line 225

  Reply: See our reply to the comment on line 225.

- Line 354: I don't follow this reasoning: Why should a 2DH model overestimating the bed load transport when wind and local morphology are with sufficiently care taken into account in the model? Therefore also the question before: is the wind playing a direct role, or is it the tide affected by the wind at the north sea? This makes a difference in how carefully local windspeed should be incorporated into the 2DH or 3D model.

  Reply: We suggest to change the text as follows:

  L354: We used a proxy to estimate the sediment transport. Based on this proxy we observed differences in the direction and the magnitude between values nearest to the bed, i.e. the lowest measurement cell, and the depth-averaged value. 2DH

models adjusted to depth averaged velocities will calculate the correct magnitude of the flow, but deviate from the actual direction. In situations when the direction of the flow in the vertical is not uniform, the calculated sediment transport on 2dH models can have the transport direction wrong. In those cases more sophisticated models are needed if one is interested in bed load transport patterns.

As is stated in the remark on L316-324 wind stress has an effect, especially on the higher elevated parts of the bar. However, external surges originated by external wind fields will contribute to changes in the peak velocities. Therefore, external wind (surge) as local wind (wind stress) is necessary for 2D as well as 3D models.

- Line 365: Sediment disposal is well documented for Western Scheldt.

  Reply: this is a general recommendation. For the Western Scheldt these locations are well known, however for other estuaries this data may be lacking or kept confidential.

- Fig. 1a: please comment on how the arrows are calculated or refer to the source.

  Reply: We will add the following text to the description of the figure: 'Patterns are derived from the calculations with the Scalwest model with a representative spring tide as boundary condition and the bathymetry of the year in question. Ebb or flood dominance is then obtained by determining the maximum depth averaged velocity in each cell.'

- Fig. 4: suggestion to ad a tidal curve of the waterlevels in the channel as well.

  Reply: Thank you. To avoid overcrowding the figure, we indicate the time of HW in each figure.

- Fig. 7: caption in the figure is hard to read.

  Reply: We will clarify it in the revised manuscript.

---

## Author Comment (AC2)

Dear authors,

I have read your manuscript with great interest. There is no doubt that this unique data set needs to be published (16! ADCPs over a period of 6 months, including a storm season). I am however not sure whether the chosen format is the best. You might consider changing the manuscript to a hydrodynamics-only paper. Leaving out the somewhat artificial way of dealing with sediment transport (a power of the velocity and neglecting waves) gives some more room for detailed analyses of the hydrodynamics. As indicated below, my main suggestions are related to the discussion part (the mass balances for cross shore and longshore). I hope these suggestions help in improving the paper.

Best regards,

Bram van Prooijen

**General remarks**

Reply: We thank the reviewer for his constructive and detailed comments.

While agreeing that a hydrodynamics paper would be valuable, our ultimate motivation is sediment transport patterns on mid-channel bars. Precisely to avoid all the intricacies and uncertainties of different sediment transport functions, we used the simple approach (common in physics) to study sediment transport patterns. As part of the PhD research of the first author, we will later study the different physical parameters and different transport predictors using numerical modelling.

We make the following suggestions (in green) to more clearly describe the goal of the paper in the section 'Introduction':

L25: For the prediction of these morphological changes, we need to have knowledge of the sediment transport patterns in intertidal areas in relation to the hydrodynamic conditions and human activities. Additionally the question is whether frequently used 1D or 2D models are sufficient to predict these changes.

L64: For this, we analysed the measured the current velocities on a spatial grid at 16 locations for a period of 6 months on a mid-channel tidal bar in the Western Scheldt estuary in the Netherlands. The results are compared to simple models developed for open and fringing flats, which serve as a starting point for understanding differences with mid-channel bars. The measured cross-shore and alongshore velocities were used as input into a simple sediment transport predictor to understand how the hydrodynamic forcing and tidal flat morphology determine the sediment balance of a mid-channel tidal bar. As the field site is macro-tidal and limited in fetch we assumed that the effect of waves on the large-scale sediment transport patterns is small in comparison with that of currents. Therefore the effect of waves on the sediment transport was omitted in this study.

L66: We compared the results to simple models developed for open and fringing flats, which serve as a starting point for understanding differences with mid-channel bars. From this comparison we will infer whether the use of 1D and 2D models is sufficient to predict morphological changes of an intertidal bar, or that more sophisticated 3D models are needed.

L69: These differences with fringing tidal flats are likely to have implications for the hydrodynamics, sediment transport, and morphological evolution of a mid-channel bar. Although flow patterns on intertidal shoals have numerically been studied (de Vet 2018, Elmilady 2020, Elmilady 2022), the specific role of the non-uniform bathymetry was not elaborated in these studies.

**Introduction**

L44-54: The flow and sediment transport on mid-channel bars is described. It is suggested in L53-54 that the flow over the flats (differences with fringing flats) will likely have an effect. This has been studied already in De Vet et al (2018), where such flows (especially during wind events) occur and where it is shown that the combination of wind and waves are crucial for sediment transport. See also the work by Elmilady et al. (e.g. 2021).

Reply: The studies mentioned look into the flow on intertidal shoals by using numerical modelling. The effect of the bathymetry on the large scale flow patterns and its effect on the sediment transport has not been further elaborated.

L55: this line needs some nuance, as there is quite some quantification by numerical modelling, but field measurements are indeed not so abundant.

Reply: We agree that many modelling results are available. However, research based on 3D modelling of mid-channels bars is rare.

Overall, the literature review could be a bit broader, like including the work of the groups from San Pablo Bay (Lacey), the Seine and other French estuaries (Grasso, Verney, le Hir), the German systems, Chinese systems like the Yangtze Estuary or Jiangsu coast, the Danish Wadden Sea? Not all need to be included, but some would embed the paper better in literature.

Reply: Thank you for these suggestions, we added several studies and propose to change the text as follows:

L56: While much understanding worldwide has been gained about flow and sediment transport on open and fringing tidal flats in estuaries, e.g. the Yangtze (Wan 2104, Zhu 2017), San Pablo Way (Allen 2019), the Gironde, Seine, and Belon estuary (Billy 2012, Florent 2019, Michel 2021), Jade, and Weser, (Gelfort 2011, Benninghoff 2017, Lefebvre 2021), and Oosterschelde (de Vet 2020), three-dimensional (3D) flow patterns on mid-channel bars are neither well known nor quantified. Measurements are sparse and costly (de Vet 2018), and morphodynamic models often use depth-averaged flow

velocities (either 1D or 2DH) for reasons of computational efficiency to calculate sediment transport and morphological change, even though flow patterns may in fact be 3D.

**Methods**

**Field site**

This section contains a lot of Dutch names that are not further used in the Results or Discussion section. You could simplify it and refer to various other papers which elaborate on the history of the Western Scheldt and tidal flats.

Reply: We propose to change the subsection 'Field site' as follows:

L85-89: As is depicted in Fig. 1(a) this mid-channel tidal bar is surrounded by multiple channels: in the south and east ebb dominant channels, and in the west and north flood dominant channels. Attached to the field site in the northeast is a spit that separates the flow from the adjacent channel in the eastern part of the field site. The mid-channel bar is fully submerged during high tide, except for the area in the southwest, which is submerged only when the water level is above 3 m NAP (NAP = Dutch ordnance datum, approximately mean sea level).

For example references would be needed to back up the last sentence "This rapid elevation decreases the intertidal area and increases the percentage of silt in the bed."

Reply: We suggest the following change to the text:

L99: Peak flow velocities decrease as the bed level becomes higher. As a result, finer sediments will settle easier, thereby increasing the percentage of silt in these areas.

**Data gathering**

This is the strong point of the paper. The set up with so many ADCPs is unique. This might be emphasized a bit more here, but also in the abstract.

Reply: thank you and we suggest the following change of the abstract:

L4: Using a unique dataset consisting of six months of current velocity data at 16 locations on a mid-channel bar, we analysed the hydrodynamics and calculated a proxy for sediment transport. Taking advantage of the large number of measurement locations, detailed flow patterns were derived. The data show that the spatially non-uniform morphology influences the hydrodynamics and sediment transport in multiple ways.

**Data Filtering**

L131-132: It is not fully clear for me why the data at VLIS and TERN were needed, as the ADCPs also record water levels.

Reply: The reason is the higher accuracy of the water level obtained at the measurement locations in the field set, i.e. locations MP11 to MP44. The method used, which is based on M2 parameters, gives an accuracy of the water level at a location of ± 0.025 m (Source: Functional requirements RWS Landelijk Meetnet Water). The RDI Workhorse ADCP (Source: Specifications Teledyne Instruments RDI Workhorse) gives an for the pressure sensor an available depth rating of 20 bar ± 0,1% or ± 0.02 m ($\rho$ = 1025 kgm$^{-3}$). The Aquadopp Current Profiles ((Source: Nortek User Guide 2008) gives the accuracy of the pressure sensor as 0.25% of the full scale (100 m), resulting in an accuracy of ± 0.25 m. In addition, both type of instruments require an accurate measurement of the local air pressure when converting pressure to water level.

L133: what does manually checked mean, what was done, what were criteria?

Reply: we suggest the following change in the paper:

L130: First, the local water level at the sensor was used to remove all data from the cells that were not fully submerged. Second, all data from the sensor starting from 1 measurement cell below the surface to the highest measurement cell was also removed. Third, all data were checked for consistency in flow magnitude and flow direction by using a threshold value of 90 counts or higher for the average amplitude signal of the different beams of the sensor. Last, all remaining data were visually checked for irregularities and discontinuities, due to e.g. turbulence or contamination. This data was manually removed. The local water level used, was obtained by interpolating the M2-tide parameters phase, amplitude, and mean sea level of the locations 'VLIS' and 'TERN' {Hydrographic Society 1987}.

**Sediment transport**

The approach to determine sediment transport is fully based on the flow; waves are not considered. This should at least be mentioned here and later on in the discussion. Waves will likely have a substantial effect on sediment transport. Why is chosen for a formulation without a critical shear stress?

Reply: We do agree on the potential importance of waves, external or locally initiated. Our motivation to ignore the effect of waves in this paper is given in the reply on the section 'Introduction'.

We chose a formulation without a critical shear stress since our approach is to determine sediment transport patterns and gradients and not the actual sediment transport. To achieve this a simple proxy is used. To assess the effect on the gradients of the threshold of motion, we could add a formulation that contains a critical shear stress:

$$Q_u = \frac{\sum_i^N \vec{u_i}(u_i^2 + v_i^2 - u_{cr}^2)}{N}$$

$$Q_v = \frac{\sum_i^N \vec{v_i}(u_i^2 + v_i^2 - u_{cr}^2)}{N}$$

where, for example, $u_{cr}$=0.2 m/s. However, due to the flood dominant character of the sediment transport the addition of a constant critical shear stress will not alter the overall pattern but will amplify the transport onto the bar during flood and weaken the effect during ebb.

We therefore suggest the following change of the text:

L146: To understand intertidal bar development, the gradients in sediment transport vectors are important. A proxy for the cumulative sediment transport at each location was calculated based on either the near-bed or the depth-averaged velocities. Sediment transport predictors for sandy suspended sediment typically have a dependence on the local flow velocity to the power 3 to 5. We left out the critical shear stress in the proxy, since this does not alter the general pattern of sediment transport, but only amplifies the flood dominant character of the system.

**Results**

**Large scale flow patterns**

L175-185 The peak flow around 50-60 min before high tide is likely a peak in longshore velocity. The peaks at 14 and 24 will likely have a cross-shore component too. This could be more emphasized here or in the discussion on the flow bending around the higher area in the west. The description in this part is somewhat unsatisfying, as it is not fully clear why these phase shift are given. Are they to be discussed in the discussion?

Reply: the phase shifts are given to indicate that the flow differs along the field site. We agree that this could indeed be more emphasized, therefore we suggest the following change of the text:

Peak flood flow is typically reached 50-60 minutes before high tide which is in agreement with the peak flow in the alongshore direction. The exceptions are locations '14' and '24'. At these two locations a clear peak during flood is missing: the flow is maximum  at the moment the sensor is  flooded and subsequently decreases. At both locations the cross-shore component has a considerable effect on the current magnitude as is shown in Fig. 1 of this reply.

[Figure]

*Figure 1 Alongshore and cross-shore current at highest locations on the field site.*

L192-197: The spring tide ebb peak in 31 and 41 is (much) more pronounced and this peak also seems to be present in 11, 12 and 32 for spring tide. This might be related to a dewatering process that is stronger during spring tides. Most spring tide flood peaks seems to be a bit later than the normal tide flood peaks.

Reply: we are inclined to agree but this is an interpretation that we are not certain about and does not directly affect our conclusions.

**Sediment transport**

L218 indicates a modulation by the spring-neap tidal cycle. This might be phrased stronger: sediment transport occurs during spring tide and almost nothing happens during neap tides. Only at 12, the alongshore component shows some changes during neap. The effect of the episodic event is indicated. This effect will in reality be much more important as waves will play a role in eroding the bed. This is relevant to note here.

Reply: We will add this to the Conclusions.

L223: "At all transects the cumulative sediment transport near the bed is smaller than the depth-averaged transport in alongshore as well as cross-shore direction." This is a direct consequence of the smaller velocity near bed than depth-averaged. You could consider to plot only the transport based on the near bed velocity, as you consider bed load. Having said that: the velocities are quite high, so suspended load might be important here. Did you check Rouse numbers?

Reply: Our interest is on the vertical velocity profile, since we are looking into the role of the local bathymetry to sediment transport. As our further research will focus on the role of dredged material to the sediment transport, specific attention in this paper is given to

the current flow near the bed, as this is mostly sand in the Western Scheldt and how this compares to the depth-averaged flows. We agree that the smaller magnitudes are due to lower near-bed velocities, but we decided to keep the transports based on the depth-averaged velocity as well, because it highlights the change in transport direction. We did not check Rouse numbers because that makes sense in a research context with sediment transport predictors.

We suggest the following change to the text:

L223: At all transects the cumulative sediment transport proxy applied to the near-bed flow velocity is smaller than that applied to depth-averaged in alongshore as well as cross-shore direction. This is a direct consequence of the smaller velocity in the lowest cell than depth-averaged. While the lower magnitude is expected, our interest is the change in direction of near-bed flow due to three-dimensionality of the flow around the bar.

**Section 4.1**

A 1D model is proposed, based on the rigid-lid approach and the subsequent volume balance. Such a model assumes that the width is uniform and that the flow is purely in one direction. A hard boundary condition was imposed at the highest point in the profile, assuming that flow cannot flow over the highest point (i.e. there is no net flow over the bar. To compare the model results with the measurements, a linear regression is proposed (L270). It confuses me a bit why such an approach is chosen, as the model is based on a volume balance, there is no calibration parameter or so. What does the values for a and b thereby mean? You could consider to plot the results and indicate that they are all below the 1:1 line. This suggests that (i) the assumed control volume (no uniform width); (ii) or the boundary condition (no flow at the highest point); or (iii) the assumption of no longshore gradients; or (iv) the assumption of a uniform water level (no friction) does not hold (a better definition of these four might be needed). By only presenting the values for a and b, it is not clear during which parts of the tide the model is performing well and when not (is there a difference between neap or spring, or between ebb or flood?). Flow over the highest point would only be of importance during emergence of that highest point. The potential causes (i-iv) could/should be unravelled in more detail. Based on the present results, it is not clear which cause is most important. It is therefore also unclear what the cross shore current drives. For example, the assumptions of a 1D model are quite violated for Transect 1. There is net cross shore flow velocity and likely also discharge? (Figure 6); there is likely a longshore gradient in longshore velocity; friction will likely play a role during dewatering. The values for a for Transect 1 are therefore much smaller than 1. This section needs to be reconsidered.

Reply: Our results show that the flow patterns are 2D/3D and we know that the mid-channel bar has a 3D morphology. As 1D models are still used in morphological studies, we question whether these models can be used on 2D/3D mid-channel bars. Therefore we compared the measured cross-shore velocities with a simple model developed for open and fringing flats, the commonly used and accepted model of Friedrichs (1996) .

This model is the simplest possible hypothesis for flow on tidal flats assuming alongshore uniformity. The results, depicted in table 1 of the paper, show that at all locations except '34' the model underestimates the cross-current velocity (factor a is less than 1).

We do agree that potential causes could further be unravelled: we know that mid-channel bar are not alongshore uniform, as stated at line 46 in the 'Introduction'. Furthermore, it is known from the field visits during this campaign that the bed-level is not uniform and differs in composition spatially and temporally. We did look into the relation between flooding of the intertidal bar and the effect on the flow, but no relation between the two was found.

Thus in order to predict flow on mid-channel bars 1D models are, as we expected, not sufficient and more sophisticated models must be used that account for the alongshore nonuniformity and possibly variations in bed stress.

We suggest the following addition to the text (in green):

L249: The cross-shore flow velocities are important for sediment transport onto and off the tidal flat. Since we know that the mid-channel bar has 3D morphology and is not alongshore uniform, we are interested whether a 1D model is suitable for such a location.

L276: adding momentum to the cross-shore current with higher velocities as a result. It is known from field visits during this campaign that the bed-level is not uniform and differs in composition spatially and temporally. We did look into the relation between flooding of the intertidal bar and the effect on the flow, but no relation between the two was found.

**Small notes**

L263: why was smoothing with a 100m window needed?

Reply: Smoothing was done to remove small ripples in the high resolution dataset.

L276: the model does not need a bed level with a constant slope (linearly changing bed level). The model can cope with a convex-up slope.

Reply: Agreed, the sentence will be removed.

L276: contraction -> contradiction?

Reply: This will be corrected in the text.

Line 279: the model assumes that the longshore flow is uniform, not zero, see also L278.

Reply: Agreed, the sentence will be removed.

**Section 4.2**

To explore the causes for the longshore current, another volume balance is considered. In this case, the balance is fully based on measured velocities and depths. If the aim is

to explore the causes for the longshore velocities, I would rather use a momentum balance. In this way, the contributions of pressure gradients, bed friction, wind stress, advection, etc could be quantified. The volume balance is now only presented for a single area and only for two tidal cycles. In L305 it is even mentioned that the method does not work for the other areas. It would be valuable to explore whether there are differences in dominating terms between spring tides and neap tides. I would also expect that wind events would have a different balance between the different terms. Furthermore, it would be interesting to compare the difference between lower and higher area for the balance between the various terms: is friction more important on the higher parts? In its present form (volume balance), it is unclear what drives the alongshore flow. A reconsideration would be needed for this section too.

Reply: We agree and already tried to solve the momentum balance, but found that due to the fact that the advective terms and the local pressure gradient were unknown, the balance could not be solved analytically without introducing a large error term. Since this error term was of the same order as some of the important terms of the momentum equation, we decided that this result could not be used. The suggestions made will be addressed in the further research.

**Section 4.3**

The title is "*Large-scale sediment transport patterns on mid-channel bars and their dependence on external forcing and morphology*". It suggests a dependence on external forcing. The sediment transport is however based on the local velocity, which is not an external forcing. Two aspects are considered (L310-313): effects of wind and effects of local morphology. An aspect that I miss is the effect of the neap-spring tidal cycle. It is mentioned a bit in the Results section. In the analysis on the effect of the wind, it is not clear whether the effect is caused by the surge or by the local wind stress. As indicated earlier, the waves will have a crucial role here too.

Reply: The effect of the spring/neap cycle is not described in the paper, since the sediment transport during neap tides is negligible. This is confirmed by the current flow during average tidal conditions and extreme conditions in figure 4, and the cumulative transport based on the proxy as shown in figure 7. To prevent possible confusion we propose to change the title into:

'Large-scale sediment transport patterns on mid-channel bars and their dependence on hydrodynamics and morphology'.

Regarding the remark about the effect of the wind, we refer to our answer on the remark regarding Lines 316-320. We motivated why waves are not considered in this paper, however, we will add a recommendation to include waves in follow-up studies to the Conclusions.

L316-320: the data set is split into two parts, based on wind speed. It is not clear which periods are in which category, nor why the Bf3 and Bf6 are chosen. Are there periods with wind >6Bf but from another direction than from the west? It would be helpful to

indicate the bands of mild conditions and storm conditions in Figure 7. It would then become clearer whether wind affects the flow (sediment transport). Figure 12 and associated text is very difficult to follow and interpret. The ebb dominance of the channel between the main bar and the spit could get some more elaboration.

Wind speeds above 6 Bf, represented in the second dataset all originated from the directions Southwest to Northwest (220° - 350°). Due to the number of periods an indication of the periods representative for 'no wind' (DS1) and 'storm' (DS2) would overcrowd the figure. Therefore we suggest to add both levels (3 Bft and 6 Bft) to Figure 3 (c) in the paper.

Reply: We suggest the following change to the text:

L316-324: To quantify the effect of wind on the peak velocities we compared the peak current velocities during situations with almost no wind (wind speed less than 3 Beaufort) with those occurring during periods with high wind speed (more than 6 Beaufort). To do so, we proceeded with the following steps. First, we divided the dataset containing peak velocities into two subsets: dataset one (DS1) containing peak velocities during wind speeds smaller than 3.3 ms$^{-1}$ (3 Beaufort), and dataset two (DS2) containing peak velocities with wind speeds larger than 10.8 ms$^{-1}$ (6 Beaufort). For both datasets the tidal range associated with the peak velocity was also registered. Second, for each measurement location a linear fit was made between the tidal range (X) and the peak velocities for ebb as well as flood (Y). Third, the obtained regression coefficients were used to calculate the peak velocities of DS2 given the tidal range in the dataset. Finally, to determine the effect of higher wind speed represented in DS2, the fraction between measured and calculated peak velocities was calculated. These are shown for flood and ebb with a logarithmic scale in Figure 12. We see the most pronounced effect on the current flow at the shallowest parts of the tidal flats, especially during flood. This strong increase of flow velocities throughout flood high up the tidal flat also explains the more episodic character of the proxy for sediment transport at the highest parts of the tidal flat. During ebb when wind and tide are in the opposite direction, there is a reduction of the peak flow, although at the most shallow locations the current flow is amplified, despite the ebb current. The mean tidal range for both periods is almost the same: 3.72 m and 3.68 m for DS1, resp. DS2. Therefore changes in the peak velocities will be mainly caused by the wind stress.

To clarify Figure 12 we propose to change the caption to:

The effect of wind on the current peak velocity for flood and ebb calculated as $\log_{10}\left(\frac{U\_\{measured\}}{U\_\{calculated\}}\right)$, with $u_{measured}$ are the measured peak velocities at wind speeds above 8 Bft, and $u_{modelled}$ are the peak velocities calculated with the coefficients derived from situations with wind speeds smaller than 3 Bft.

The explanation in L330 is a bit short. Can you explain more about the dewatering of the shoal via that channel?

Reply: The ebb dominant character of these two locations could possibly be explained by dewatering of the shoal and the flow coming from the ebb dominant part of the channel 'Schaar van Spijkerplaat'. The latter contributes to the current flow at these locations as long as the northern sand spit is submerged, which is reached at water levels above -2 m NAP.

Figure 2 shows a clear difference in bedform patterns. Are these differences to be considered too?

Reply: Several distinct and different bedform patterns are visible in the high resolution bathymetry. These differences were taken into consideration while analysing the ebb and flood dominant character of the area. The patterns show that the western part, as well as the 'Schaar van Spijkerplaat' is flood dominant. At the northern sand spit the bed forms are orientated to the ebb direction which confirms the ebb-dominant character of the area.

**Conclusions**

343-350: The assumption is made that it is commonly assumed that cross-shore velocities are the only driver for filling/emptying a tidal flat. This is maybe somewhat overstated. Other authors (e.g. de Vet et al) also indicated already the more complex flow patterns on shoals.

Reply: There have been studies carried out on flow patterns on mid-channel bars (de Vet 2018, Wang 2019, Elmilady 2020), However, the number of such studies is limited. These studies are referred to in the previous sections, or suggested to add in this document. We therefore choose to omit these references in de conclusion sections. However, we propose to change 'common' into 'often-used'.

L344: It is not convincingly shown that 2DH models overestimate bed load transport. The argument given in the next sentence mention a representative time period. These seem two different aspects: (i) is a 2DH model able to represent the flow?; (ii) can a representative period be used for upscaling? A statement about the validity of numerical models would need more elaboration.

Reply: We suggest the text as follows:

L354: We used a proxy to estimate the sediment transport. Based on this proxy we observed differences in the direction and the magnitude between values nearest to the bed, i.e. the lowest measurement cell, and the depth-averaged value. Although magnitude differences in the vertical profile can be solved by a 2DH model, the question is if and how 2DH models differences in direction over the vertical profile can be solved or that more sophisticated models are needed.

The impact of waves is not considered in this study. It should at least be mentioned somewhere. How would the wave impact change the conclusions?

Reply: we refer to the previous remark and the answer and propose to change the text as:

L365: For further studies that take morphological changes and sediment transport into account it would be useful to identify possible sediment sources, including locations of disposal sites and quantities of disposed sediment. It is advised to look into the effect of waves on the large sediment transport patterns, numerically as well as through measurements. Measurements are also advised on the sediment transport in suspension and on the bed. Besides providing valuable information, data on sediment transport and local wave information provide valuable information to validate model results.

L370: this is a bit demotivating for almost all readers, especially as last sentence. Even 16 ADCPs are not enough to identify the main flow patterns? Most readers will likely have much less instruments. It would be valuable to indicate what would be possible with less instruments. Would you place them in a transect or in a cross?

Reply: we agree, though not with disappointment but intrigued because it also surprised us. The main issue is that flow depends on local bathymetry, which has variations at different spatial scales. A fundamental question is how representative the flow at one location is for a larger area, and at which scale the flows needs to be measured. It is recommended to always try to cover the spatial variations of the area you study. If the system is alongshore uniform, a cross-shore array will do. If main spatial changes are alongshore, an alongshore array is needed. If the system is fully 2D, it should be either a cross or an array as used here.

We suggest to change the text as follows:

Elucidating these processes and assessing how important the intricate patterns are for sediment transport requires a combination of numerical modelling and measurements that cover the spatial variations of the study area. If this area is alongshore uniform, a cross-shore line-up is sufficient. When the main spatial changes are dominated by the alongshore velocity component, a line-up in the alongshore direction is needed. When the study area is fully 2D, the instruments may be placed in a cross line-up, or as an array as used in this study.